# Factors Influencing Physician Decision Making to Attempt Advanced Resuscitation in Asystolic Out-of-Hospital Cardiac Arrest

**DOI:** 10.3390/ijerph18168323

**Published:** 2021-08-06

**Authors:** Charles Payot, Christophe A. Fehlmann, Laurent Suppan, Marc Niquille, Christelle Lardi, François P. Sarasin, Robert Larribau

**Affiliations:** 1Division of Emergency Medicine, Department of Anaesthesiology, Clinical Pharmacology, Intensive Care and Emergency Medicine, Geneva University Hospitals, 1211 Geneva, Switzerland; charles.payot@hcuge.ch (C.P.); christophe.fehlmann@hcuge.ch (C.A.F.); laurent.suppan@hcuge.ch (L.S.); marc.niquille@hcuge.ch (M.N.); francois.sarasin@hcuge.ch (F.P.S.); 2School of Epidemiology and Public Health, University of Ottawa, Ottawa, ON K1G 5Z3, Canada; 3Emergency Medicine, Research Group, Ottawa Hospital Research Institute, Ottawa, ON K1Y 4E9, Canada; 4University Center of Legal Medicine (CURML), Geneva University Hospitals, 1211 Geneva, Switzerland; christelle.lardi@hcuge.ch

**Keywords:** out-of-hospital cardiac arrest, emergency physician, medical decision, asystole, advanced life support, Charlson comorbidity index, emergency medical service, prehospital emergency

## Abstract

The objective of this study was to identify the key elements used by prehospital emergency physicians (EP) to decide whether or not to attempt advanced life support (ALS) in asystolic out-of-hospital cardiac arrest (OHCA). From 1 January 2009 to 1 January 2017, all adult victims of asystolic OHCA in Geneva, Switzerland, were retrospectively included. Patients with signs of “obvious death” or with a Do-Not-Attempt-Resuscitation order were excluded. Patients were categorized as having received ALS if this was mentioned in the medical record, or, failing that, if at least one dose of adrenaline had been administered during cardiopulmonary resuscitation (CPR). Prognostic factors known at the time of EP’s decision were included in a multivariable logistic regression model. Included were 784 patients. Factors favourably influencing the decision to provide ALS were witnessed OHCA (OR = 2.14, 95% CI: 1.43–3.20) and bystander CPR (OR = 4.10, 95% CI: 2.28–7.39). Traumatic aetiology (OR = 0.04, 95% CI: 0.02–0.08), age > 80 years (OR = 0.14, 95% CI: 0.09–0.24) and a Charlson comorbidity index greater than 5 (OR = 0.12, 95% CI: 0.06–0.27) were the factors most strongly associated with the decision not to attempt ALS. Factors influencing the EP’s decision to attempt ALS in asystolic OHCA are the relatively young age of the patients, few comorbidities, presumed medical aetiology, witnessed OHCA and bystander CPR.

## 1. Introduction

When a patient suffers an out-of-hospital cardiac arrest (OHCA), healthcare professionals providing advanced life support (ALS) measures during the prehospital phase must take crucial decisions whilst in the field [1], notably when to provide prehospital ALS, which presents a challenge [2,3].

First, these professionals must decide whether or not to start resuscitation when it has not already been started by a bystander [3]. Second, elements related to the circumstances of the OHCA and the patient’s wishes and clinical condition are collected, which may lead to a decision to stop resuscitation manoeuvres early [4]. Finally, in the case of prolonged asystole with more than 20 min of cardiopulmonary resuscitation (CPR) without a treatable cause, they must decide whether or not it is possible to transport with on-going CPR [5,6,7].

Normally, without overt clinical signs of irreversible death (e.g., postmortem lividity, rigor mortis, decapitation, decomposition), or (in Europe) a valid Do Not Attempt Resuscitation (DNAR) order, advanced life support (ALS) providers must initiate CPR [3]. There are local recommendations to guide emergency medical services (EMS) as to whether or not to start resuscitation for OHCA, but international guidelines have not yet been issued [8,9,10]. Some studies show that early CPR is less often initiated when the OHCA is not witnessed or when asystole is present, even in the absence of obvious signs of death or a DNAR order [11,12,13].

To avoid the futile transport of OHCA patients to hospital with virtually no chance of survival, the termination of resuscitation (ToR) rules are currently used as a guide for discontinuing CPR in the field. For ALS providers, these rules are based on the absence of five factors: OHCA witnessed by EMS personnel, shockable rhythm, return of spontaneous circulation (ROSC), OHCA witnessed by a bystander and bystander-administered CPR [14]. These ToR rules have been adapted to local conditions in different countries [15].

In Europe, and especially in the Franco-German EMS, prehospital EPs constitute the highest level of advanced life support providers, and they intervene with the ambulance team in the field [16,17]. In OHCA situations, these EPs provide advanced care to patients and make medical decisions (e.g., decision to stop CPR and declare death in the field) [18].

When in the field, the EPs may decide not to perform advanced resuscitation or to stop resuscitation early where an OHCA has occurred, especially when the initial rhythm is asystole. The factors that influence their decision to perform or not to perform advanced resuscitation in the field are not precisely known.

The objective of this study was to identify the key elements used by such EPs when deciding whether or not to attempt ALS manoeuvres in adult victims of OHCA whose initial rhythm was asystole.

## 2. Materials and Methods

This report follows the Strengthening the Reporting of Observational Studies in Epidemiology (STROBE) statement guidelines for reporting cohort studies [19].

### 2.1. Study Design and Setting

This study was based on a retrospective analysis of the OHCA register in Geneva, Switzerland. It was approved by the Geneva Cantonal Commission for Research and Ethics (Identification No: 12-208-R). Patient consent was waived by this committee. Medical records were all computerized.

At the end of 2016, the permanent population residing in the Canton of Geneva was of almost 500,000 inhabitants; during daytime, this figure is increased by around 100,000 daily commuters from France and other Swiss cantons. The emergency medical communication centre (EMCC) centralizes all requests related to prehospital medical emergencies, including OHCA. In Geneva’s EMCC, dispatchers have a paramedical or nursing background and coordinate emergency mobile units remotely by sending appropriate response teams.

In the Canton of Geneva, the EMS is two-tiered (or three) with different medical levels and skill sets. The first level is made up of emergency ambulances, staffed by two ALS-trained paramedics. There are fifteen ambulance bases scattered throughout the Canton of Geneva that operate according to the proximity of the base to the scene of the emergency. The second level is made up of a Mobile Emergency and Resuscitation Service or *SMUR (Service Mobile d’Urgence et de Réanimation)*, a light vehicle that operates with a certified paramedic and an EP in training (junior or intermediate) whose background and level of expertise can vary, but who has at least 2 years of experience [20]. If necessary, specialist senior EPs are available 24 h a day to intervene on-site (third level), for example, in the event of a difficult intubation, a refractory OHCA or if the junior EP is already busy with another emergency event [21]. To improve their skills and knowledge, these junior EPs follow a number of internal courses, including resuscitation simulation. Official training courses such as the Advanced Cardiovascular Life Support (ACLS) are not mandatory but highly recommended. Senior EPs review all EMS intervention reports daily for teaching and quality control purposes.

### 2.2. Prehospital Management of OHCA and Decisions

There are three decision points during the management of an OHCA in Geneva (Figure 1). Decisions at points 2 and 3 are made by the attending EP. The EP has no specific guidelines imposed on him/her when making decisions not to start or to withhold CPR. The ToR guidelines for ALS are not applied in Geneva regarding decision point number 3 (before transport).

### 2.3. Study Population

The medical records of all patients were screened for whom an emergency call for OHCA was received by the EMCC between 1 January 2009 and 1 January 2017. Patients were eligible for inclusion if they presented a confirmed OHCA case in the Canton of Geneva and had been taken care of by the *SMUR*. Patients were then excluded if they were not in asystole, presented obvious signs of death (post mortem lividity, rigor mortis or life-incompatible injury), had a DNAR order, or were younger than 16.

### 2.4. Variables

The outcome was the medical decision to perform an advanced resuscitation. Therefore, a patient in whom we decided not to start ALS or where we decided to stop very early (after receiving information regarding the patient’s wishes, clinical condition and the circumstances of the OHCA) were considered not to have the outcome.

Indeed, early administration of adrenaline is recommended by the ACLS guidelines [6]. The medical decision not to resuscitate was defined as the mention in the intervention report, of abstention from CPR, or nonuse of intravenous adrenaline in the absence of return of spontaneous circulation (ROSC). Therefore, we considered that the EP did not intend to provide advanced resuscitation if resuscitation was stopped prior to the administration of the first dose of adrenaline.

Factors that may influence the medical decision to perform advanced resuscitation were identified on the basis of a conceptual framework. These factors were included only if they were known by the EP at the time the decision was made. For patients, the factors included were sex, age, comorbidities and the presumed aetiology of OHCA. The presumed aetiologies were classified (according to Utstein 2015) as medical (cardiac and noncardiac), traumatic, asphyxia and unknown [22]. Patient comorbidities were defined by the Charlson comorbidity index (CCI), which was collected retrospectively in a prehospital chart review [23]. Concerning the circumstances of the OHCA, the presence (or absence) of witnesses at the time of the cardiac arrest and the CPR performed (or not) by bystanders was also considered.

For ALS providers, we took into account the response time of the first team in the field and the gender and experience of the lead prehospital physician. EP’s experience was defined as “junior” (less than 5 years of medical residency), “intermediate” (specialist certification or more than 5 years medical residency, without a supervisory role) or “senior” (prehospital specialist EP with a supervisory role in the prehospital unit).

### 2.5. Statistical Analysis

Statistical analysis was performed using STATA version 16 (Stata Corporation, College Station, TX, USA).

Patients’ characteristics are expressed as means ± standard deviation for continuous variables and as frequency and relative percentage for categorical variables. Comparisons between groups were performed using Student *t*-test or chi-squared test, as appropriate. Bivariate logistic regression analysis was used to study the different associations. We then built a multivariable logistic regression model, including our prespecified factors, and reporting the full model, even if some factors were not statistically significant. Continuous variables were categorized if the linearity of the log-odds was not respected, based on previously used categories. Collinear variables were excluded from the model. The “goodness of fit” to the model was checked globally using the Hosmer–Lemeshow test.

We performed three prespecified sensitivity analyses. In the first case, the outcome was defined only based on the mention of “abstention of resuscitation”, without taking into account adrenaline use. In the second case, the cutoff used to determine the “intermediate” level of experience was lowered from 5 to 3 years. Finally, we excluded patients with a presumed nonmedical OHCA aetiology.

Missing values were reported as such and coded as “unknown”; no multiple imputations were performed. Based on the estimation that the decision not to perform resuscitation occurs in about 20% of OHCA, 500 patients would have been needed to adjust for 10 potential predictors. Based on the estimated average of 80 potential patients a year, an 8 year period was considered for this study. For all tests, a two-sided *p*-value below 0.05 was considered significant.

### 2.6. Patients and Public Involvement

Patients and the public were not involved in the design or planning of the study.

## 3. Results

There were 2981 OHCA patients considered for inclusion. The two most frequent exclusion criteria were the presence of obvious signs of death (*n* = 1319) and an initial rhythm other than asystole (*n* = 711). Finally, 784 patients were included in the analysis (Figure 2).

The exact arrival times of the ambulance crew and the *SMUR* team on-site were reported in 644 (82.1%) of the 784 OHCA cases included. The average response time of the EMS was 9′32″ (±4′00″). The first team on-site was the ambulance crew in 527 (81.8%) OHCA cases, while it was the *SMUR* team in 74 (11.5%) OHCA cases. In 43 (6.7%) OHCA cases, both teams arrived on-site at the same time. In the 74 situations where the *SMUR* team arrived on the scene before the ambulance crew, the *SMUR* team arrived on average 2’08″ (±3’00″) before the ambulance crew. The rate of patients resuscitated was the same regardless of which team arrived first on-site (*p* = 0.225).

Table 1 presents the patients’ characteristics. Out of the 784 patients, a decision to not attempt advanced resuscitation was taken for 185 (23.6%) (Figure 2). These patients were older than those in whom resuscitation was initiated, had more comorbidities, and their presumed OHCA aetiology was more frequently traumatic or unknown. Their OHCA was less likely to have been witnessed and CPR was provided less often prior to the arrival of the EMS team. There was no difference associated with either the sex or the experience of the lead physician.

Table 2 presents the unadjusted and adjusted associations. In the univariate analysis, the factors associated with the decision to attempt advanced resuscitation were sex (male), younger age, few comorbidities, presumed medical aetiology, witnessed arrest and bystander CPR. In our multivariable model, factors favourably influencing the decision to provide ALS by the prehospital EP were young age, low CCI, presumed medical aetiology, witnessed out-of-hospital cardiac arrest and bystander CPR.

The Hosmer–Lemeshow test (Chi2 = 8.92, *p* = 0.349) validates the multivariate logistic regression model goodness of fit. Finally, the three prespecified sensitivity analyses did not substantially affect the results.

## 4. Discussion

In OHCA cases with asystole as initial rhythm, we observed that age less than 65 years, absence of comorbidities, presumed medical aetiology, witnessed OHCA and bystander CPR were independent predictors that favourably influenced the prehospital EP’s decision to attempt advanced resuscitation.

These results are consistent in many respects with a recent Austrian study which showed that resuscitation is unlikely to be initiated by the EP if patients are in asystole, elderly, have significant comorbidities such as malignancy or have not received CPR prior to the EP’s arrival [24]. In Geneva, there was no association with EMS response time, but this may be explained by the very short response times in Geneva, whose area is more limited than that of the Graz region in Austria. The variable “first on the scene” was not included in the multivariable model because the differences in response time between the *SMUR* team and the ambulance crew are extremely small and not significant. These differences in response times between the EP and the ambulance crew were not published in the Austrian study and it is therefore difficult to compare with them in this respect. The very short response time of the Geneva EMS may also explain why the Geneva EP attempts ALS in 76.4% of asystolic OHCAs, whereas the Austrian EP only attempts ALS in 62% of asystolic OHCAs.

Another study also showed that old age, previous poor health and lack of CPR initiation influenced the EP’s decision not to initiate advanced resuscitation [25]. These results are also consistent with current knowledge about prognostic factors in OHCA. Advanced age [26], a high number of comorbidities [27], a nonwitnessed OHCA, absence of bystander administered CPR [28] and traumatic aetiology are indeed well known to reduce survival rates after an OHCA [29]. These medical decisions not to attempt advanced resuscitation, made on the basis of knowledge of poor prognostic factors, are also consistent with decisions made when a physician believes that the prognosis is very poor and that further treatment would be futile [30]. In the emergency department, the main factors influencing the decision not to provide resuscitative care are old age and previous severe functional limitations [31]. It is therefore reassuring that EPs are making decisions consistent with knowledge of the prognostic factors associated with OHCA.

In this study, we measured the CCI retrospectively for each patient, based on a systematic review of medical records. Although this may be considered a limitation, the influence of patients’ comorbidities on the medical decision to attempt ALS has been reported infrequently in the literature. However, we found that the presence of comorbidities does seem to influence this decision. Ideally, comorbidities should be systematically documented, as well as their relationship to medical decision making. Another way to achieve this could be the prospective integration of a frailty score, such as the clinical frailty scale (CFS) developed by Rockwood [32]. This tool, which has been shown to be relevant for the limitation of life-sustaining treatment in the ICU [33], could be included as a core variable in the Utstein resuscitation registry template [22].

We deliberately limited the scope of this study to asystole situations, as we assumed that advanced resuscitation was routinely provided when a shockable rhythm or pulseless electrical activity was noted on the EPs arrival at the emergency site. This may be debatable, especially in situations of pulseless electrical activity, which may not be resuscitated in real life. However, the recent Austrian study shows that when an electrical rhythm is present on the electrocardiogram, the EP tends to systematically provide advanced life support [24].

When the EP is confronted with an OHCA, a time delay is required to gather the information necessary to make a decision. When BLS is in progress, the EP will not interrupt it, and may even start ALS, whilst looking for futility criteria at the same time. This is rarely documented in the medical records and is not one of the variables to be collected in the Utstein resuscitation registry template [22]. For this reason, we chose the criterion of “nonuse of intravenous epinephrine in asystole” to ascertain the lack of intent to provide ALS as it can easily and reliably be measured retrospectively. The intention to initiate ALS and the intention to transport under ongoing CPR, after ALS has been delivered at the emergency site, are two temporally successive decision points. Advanced ToR rules have been proposed to avoid patient transport whilst under continuous CPR, especially when there is no EP present who can intervene in the field and decide to stop ALS [14]. There are no rules on which advanced care providers can base a decision to withhold giving ALS, and recognition of early criteria for futility, such as postmortem lividity, can be difficult. Knowledge of the factors on which EPs make these decisions and the relevance of these criteria are therefore the first steps in developing decision support tools in this area. These tools can be very useful in helping EPs when making a decision not to attempt ALS in futile situations [34].

The standardized description of the OHCA care process (Figure 1) in a business process model and notation (BPMN) [35] makes it possible to highlight the three successive decision points of the prehospital phase in Geneva. The second decision point (decision to attempt ALS) is very important because the proportion of survivors at discharge (or at 30 days) are generally measured against this decision point, so the better the selection of patients for whom ALS is performed, the better the final survival rate. Although this variable is not reported in the Utstein resuscitation registry template, the rate of ALS is likely to be lower when an EP is dispatched to the field to make this decision. As a result, the survival rate at discharge of a patient who received ALS is likely to be better in these systems than in systems where this decision cannot be made in the field. This has already been demonstrated with the implementation of the ToR rules for the third decision point [36].

In the prehospital setting, medical decisions to withhold and withdraw care are therefore common [37]. Unfortunately, these decisions are often made by EPs alone in the field [25]. A previous study also showed that less-trained clinicians tended to forego care in emergency departments more often than physicians with more years of training [38]. However, in our study, we did not observe any difference in the number of years of medical residency regarding decision making. This could be explained by our setup, where senior physicians readily support EPs in training, either directly in the field or by phone. EPs in training can therefore always count on the support of a senior physician when making the decision as to whether or not advanced resuscitation should be attempted. Another explanation could be the ethics and decision-making training provided early-on during pregraduate studies and medical residency in Switzerland.

This study has several limitations. Indeed, it is a retrospective and monocentric study, with all the limitations associated with its study design. In particular, the retrospective design limits the number of factors that can be studied, as only those collected for the medical record documentation could be included. Prospective studies, or even qualitative studies with residents, may help to identify additional factors influencing medical decision making. Moreover, the Canton of Geneva is small and essentially urban, so the generalisation is limited to similar territories, in particular with regard to response times. Another limitation is related to the fact that the family’s request to resuscitate the patient (or not) was not systematically recorded in the medical records. This variable could therefore not be included in the model. Furthermore, the local Geneva population includes many different ethnicities and religions, with diverse perspectives on resuscitation that may have an impact on the families’ requests. Finally, it is an emergency medical system where ALS is provided by paramedics and, in addition, where an EP can be dispatched to the field. Few emergency medical systems are comparable to Geneva’s emergency medical systems. The observation period is eight years; it is possible that changes in decision making during this period are not taken into account. We should also note that the criterion of “no use of intravenous (or intraosseous) epinephrine in situations” as a definition of no intention to provide ALS may be questionable. A bias may have been created by the fact that EPs reported more comorbidities in the medical records when making the decision not to initiate ALS, in an attempt to justify it.

The results of this study are in line with the new guidelines on the ethics of resus-citation and end-of-life decisions, published in 2021 [39]. Indeed, this study shows that EPs are unlikely to base their decisions solely on isolated clinical signs or isolated markers of poor prognosis. They also take into account, for example, age and comorbidities. The ToR rules, developed for emergency medical systems that do not send EPs to the field, are not applied in Geneva. The 2021 guidelines on the ethics of resuscitation and end-of-life decisions recommend “that none of the existing ToR rules should be the sole determinant of when to discontinue resuscitation”. In all probability, Geneva EPs take the criteria defined in the ToR rules into account when making a decision not to transport a patient with ongoing CPR. However, in this work, we studied the decision to attempt (or not) resuscitation (Figure 1, decision point 2), not the decision to transport (or not) the patient with ongoing CPR (Figure 1, decision point 3).

These results deserve to be confronted with a future study that would prospectively measure whether EPs who make these decisions do so both ethically and appropriately. Patient survival (at hospital discharge or at 30 days), based on the decision criteria identified here, and where ALS care was provided, should be measured, as has already been done for ToR rules in 2006 [14].

## 5. Conclusions

An EP attempts ALS in more than three quarters of OHCA cases with an initial rhythm of asystole. The factors that favourably influence the EP’s decision to attempt ALS are age under 65 years, absence of comorbidities, presumed medical aetiology, witnessed OHCA and bystander CPR. The factors most strongly associated with the medical decision not to attempt ALS are traumatic aetiology, aged over 80 years and a CCI greater than 5.

The medical decision whether or not to attempt ALS should be routinely reported in the Utstein resuscitation registry OHCA template, at least in emergency medical systems using a prehospital EP.

Future studies are needed to prospectively measure whether EPs who make these decisions in the field are doing so both ethically and appropriately.

## Figures and Tables

**Figure 1 ijerph-18-08323-f001:**
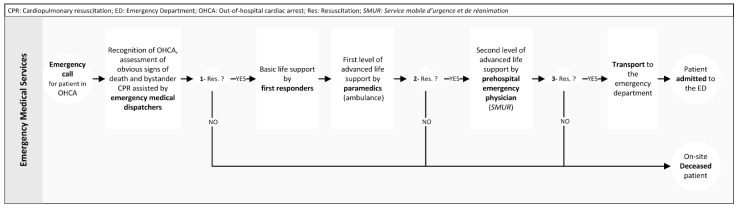
Geneva process for the management of out-of-hospital cardiac arrest and decision points.

**Figure 2 ijerph-18-08323-f002:**
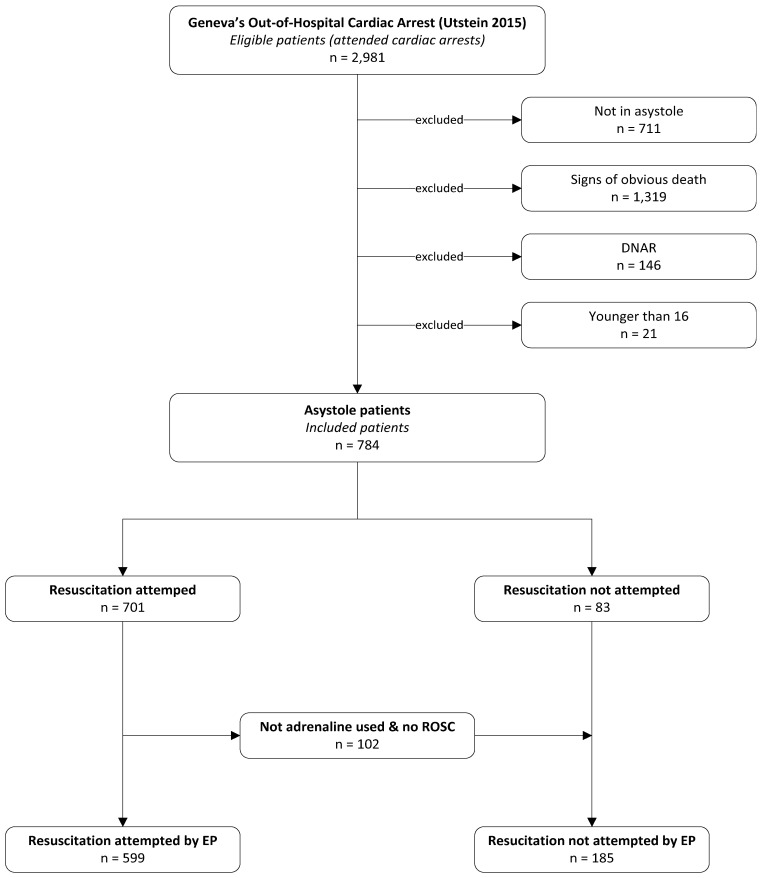
Flowchart of the study.

**Table 1 ijerph-18-08323-t001:** Patient characteristics.

Variables	Total	Resuscitation Attempted by EP	Resuscitation Not Attempted by EP	*p*-Value ^2^
	*n* = 784	*n* = 599	*n* = 185	
Patient’s sex (male), *n* (% ^1^)	496 (63.3)	392 (65.4)	104 (56.2)	0.023
Patient’s age (years), mean ± SD	66.9 (±18.1)	64.6 (±17.6)	74.4 (±17.8)	<0.001
Charlson comorbidity index, *n* (% ^1^)				
*0*	427 (54.5)	341 (56.9)	86 (46.5)	0.001
*1–2*	277 (35.3)	210 (35.1)	67 (36.2)	
*3–4*	38 (4.9)	26 (4.3)	12 (6.5)	
*5+*	42 (5.4)	22 (3.7)	20 (10.8)	
Presumed aetiology, *n* (% ^1^)				
*Medical (cardiac and noncardiac)*	256 (32.7)	218 (36.4)	38 (20.5)	0.001
*Trauma*	51 (6.5)	20 (3.3)	31 (16.8)	
*Asphyxiation (external causes)*	40 (5.1)	35 (5.8)	5 (2.7)	
*Unknown*	437 (55.8)	326 (54.4)	111 (60.0)	
Witnessed arrest (yes), *n* (% ^1^)	415 (52.9)	337 (56.3)	78 (42.2)	0.001
Bystander CPR (yes), *n* (% ^1^)	187 (23.9)	169 (28.2)	18 (9.7)	<0.001
EMS response time (min), mean ± SD	9.2 (±4.3)	9.2 (±4.1)	9.3 (±4.8)	0.869
EMS response time, *n* (% ^1^)				
*0–2 min*	24 (3.1)	14 (2.3)	10 (5.4)	0.305
*2–6 min*	100 (12.8)	79 (76.4)	21 (11.4)	
*6–9 min*	274 (35.0)	212 (35.4)	62 (33.5)	
*9–12 min*	200 (25.5)	149 (24.9)	51 (27.6)	
*12–25 min*	128 (16.3)	98 (16.4)	30 (16.2)	
*>25 min*	9 (1.2)	6 (1.0)	3 (1.6)	
*Missing*	49 (6.4)	41 (6.8)	8 (4.3)	
Lead physician’s sex (male), *n* (% ^1^)	488 (62.2)	373 (62.3)	115 (62.2)	0.979
Lead physician’s experience, *n* (% ^1^)				
*Junior*	456 (58.2)	353 (58.9)	103 (55.7)	0.282
*Intermediate*	220 (28.1)	160 (26.7)	60 (32.4)	
*Senior*	108 (13.8)	86 (14.4)	22 (11.9)	

^1^ Percentages may not total 100 due to rounding. All variables given as numbers (column percentages in parenthesis); ^2^ Based on Student *t*-test or chi-squared test as appropriate; SD: standard deviation; CPR: cardiopulmonary resuscitation; EMS: emergency medical system.

**Table 2 ijerph-18-08323-t002:** Univariable and multivariable logistic regression.

Variables	Unadjusted OR (95% CI)	Adjusted OR (95% CI)
Patient’s sex (male)	1.47 (1.05–2.06)	1.15 (0.77–1.72)
Patient’s age (years)		
*18–64*	Ref.	Ref.
*65–79*	0.69 (0.44–1.07)	0.52 (0.32–0.89)
≥*80*	0.24 (0.16–0.36)	0.14 (0.09–0.24)
Charlson comorbidity index		
*0*	Ref.	Ref.
*1–2*	0.79 (0.55–1.14)	0.57 (0.36–0.89)
*3–4*	0.55 (0.26–1.13)	0.41 (0.17–0.95)
*5+*	0.28 (0.14–0.53)	0.12 (0.06–0.27)
Presumed aetiology		
*Medical (cardiac and noncardiac)*	Ref.	Ref.
*Trauma*	0.11 (0.06–0.22)	0.04 (0.02–0.08)
*Asphyxiation (external causes)*	1.22 (0.45–3.31)	0.75 (0.25–2.26)
*Unknown*	0.51 (0.34–0.77)	0.55 (0.35–0.87)
Witnessed arrest (yes)	1.76 (1.26–2.46)	2.14 (1.43–3.20)
Bystander CPR (yes)	3.65 (2.17–6.12)	4.10 (2.28–7.39)
EMS response time		
*0–2 min*	Ref.	Ref.
*2–6 min*	2.69 (1.05–6.90)	2.00 (0.63–6.35)
*6–9 min*	2.44 (1.03–5.70)	1.91 (0.65–5.56)
*9–12 min*	2.09 (0.87–4.90)	1.48 (0.5–4.40)
*12–25 min*	2.33 (0.94–5.70)	1.86 (0.6–5.78)
*>25 min*	1.43 (0.29–7.10)	1.14 (0.16–8.03)
*Missing*	3.66 (1.21–11.10)	1.95 (0.51–7.51)
Lead physician’s sex (male)	1.00 (0.72–1.41)	0.99 (0.67–1.47)
Lead physician’s experience		
*Junior*	Ref.	Ref.
*Intermediate*	0.78 (0.54–1.13)	0.83 (0.54–1.28)
*Senior*	1.14 (0.68–1.91)	1.31 (0.71–2.41)

Logistic regression: OR > 1: in favour of performing advanced life support; OR: odds ratios; CI: confidence interval; CPR: cardiopulmonary resuscitation; EMS: emergency medical system; EP: emergency physician.

## Data Availability

The data presented in this study are available on request from the corresponding author.

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
