# Peer review of "Factors Influencing Physician Decision Making to Attempt Advanced Resuscitation in Asystolic Out-of-Hospital Cardiac Arrest"

_ijerph, 2021, doi:10.3390/ijerph18168323_

Round 1

Reviewer 1 Report

Thank you for the opportunity to review the manuscript by Payot and colleagues. The topic is moderately interesting, the study well designed and conducted. The manuscript is linear and easy to read and the results are presented straightforward. I only can suggest few minor points that could be useful to increase the straight of the manuscript.

All of the essential informations are present in the manuscripts, yet I would like to know how many patients in the cohort were included in pre-hospital interventional trials, as this could have motivated/required the teams to perform resuscitation manouvers in the pre-hospital setting. 

Secondarily, i believe the data presented in the table 1 do not need to be presented also in the text (such as the age or sex).

Another factor that might be determinant in the decision to attempt or not resuscitation manouvers is the family request, that in certain areas is reflected in the ethnicity of the patients. Even if the specific area this information might be less important, the general limitation in building similar studies might be put in evidence, or developed within the discussion.

Considering the very short response time, someone might argue that programs involving mechanical CPR and eCPR could increase the number of patients eligible for organ donation. I would like to know if any specific protocol is active in the ares, despite not being directly associated with the study outcome.

Maybe since the timeline of the study expanded over three different CPR guideline, an interesting separation might be obtained by different time clusters (07-10/10-15/after 15), or avoided based on something.

Author Response

Response to Reviewer 1 Comments

Point 1: Thank you for the opportunity to review the manuscript by Payot and colleagues. The topic is moderately interesting, the study well designed and conducted. The manuscript is linear and easy to read and the results are presented straightforward. I only can suggest few minor points that could be useful to increase the straight of the manuscript.

Response 1: Thank you very much for this comment.

Point 2: All of the essential informations are present in the manuscripts, yet I would like to know how many patients in the cohort were included in pre-hospital interventional trials, as this could have motivated/required the teams to perform resuscitation manouvers in the pre-hospital setting.

Response 2: Thank you for this pertinent question.  During the study period, there was no prehospital interventional trial performed in our unit. Only observational studies (not interventional studies) including data related to the Geneva Out-of-Hospital Cardiac Arrest (OHCA) Registry were conducted. Examples of observational studies using data from the Geneva OHCA Registry: manuscript references 17 and 20. Therefore, the decision to perform resuscitation manoeuvres should not have been influenced by that.

Point 3: Secondarily, i believe the data presented in the table 1 do not need to be presented also in the text (such as the age or sex).

Response 3: Indeed, this is redundant and we have adapted the text in the "result" section accordingly. The sentences (page 6) "The patients were mostly men. The mean age was 66.9 (SD=18.1) years. More than half of the patients did not have any known comorbidity. Half of the OHCA were of an unknown presumed aetiology and an equivalent proportion occurred in the presence of witnesses. CPR manoeuvres were performed by bystanders in less than one out of four patients." have been removed from the text in the results section. However, we considered that the description of the differences between the two groups was still worth presenting in the text and therefore kept this section

Point 4: Another factor that might be determinant in the decision to attempt or not resuscitation manouvers is the family request, that in certain areas is reflected in the ethnicity of the patients. Even if the specific area this information might be less important, the general limitation in building similar studies might be put in evidence, or developed within the discussion.

Response 4: Indeed, this is an important issue, and we thank you for mentioning it. When there is no Do Not Resuscitate order signed by the patient immediately available, the patient's wish is reported by the family. Sometimes it is the family's request (related to the ethnic or religious context) more than the patient's wish that is reported. In Geneva, the family's request probably plays a role in the medical decision not to attempt resuscitation or to stop it quickly. Unfortunately this variable is not clearly mentioned in the medical records. This is indeed a limitation of the study. Nevertheless, the population of the Geneva region includes different ethnicities and religions (>40% non-Swiss inhabitants in the canton of Geneva), with probably very different perspectives regarding the decision to resuscitate. We have included the following sentence in the "limitation" paragraph of the discussion section: "Another limitation is related to the fact that the family's request to resuscitate the patient (or not) was not systematically recorded in the medical records. This variable could therefore not be included in the model. Furthermore, the local Geneva population includes many different ethnicities and religions, with diverse perspectives on resuscitation that may have an impact on the families' requests."

Point 5:  Considering the very short response time, someone might argue that programs involving mechanical CPR and eCPR could increase the number of patients eligible for organ donation. I would like to know if any specific protocol is active in the ares, despite not being directly associated with the study outcome.

Response 5: Thank you for this question. Indeed, response times are short in Geneva. Mechanical CPR has been practiced since 2010 and allows, in some cases, the transport of the patient with ongoing CPR. e-CPR is also practiced, but only in an in-hospital setting (not in prehospital setting) since 2011, and after transport with ongoing CPR. e-CPR is performed in the hospital only in situations with very precise selection criteria (patient < 55 years, cardiac arrest in the presence of a professional witness, no-flow time = 0, time between collapse and cannulation < 1 hour, etc.). These measures (mechanical CPR and e-CPR) are never implemented for the purpose of organ donation. In Geneva, there is no "non-heart-beating organ donation" programme. Only the patient is considered when a resuscitation is performed and therefore we do not have any specific active protocol for the purpose of non-heart-beating organ donation.

Point 6:  Maybe since the timeline of the study expanded over three different CPR guideline, an interesting separation might be obtained by different time clusters (07-10/10-15/after 15), or avoided based on something..

Response 6: Indeed, the guidelines changed three times during the study period. The duration of the study (8 years) was mentioned as a limitation. The guidelines did change during the study period, but mainly regarding treatment options during resuscitation, and this may have had an impact on patient outcome. In contrast, there has been very little change in guidelines regarding the decision to perform (or not) a resuscitation. These changes in guidelines have therefore probably had little effect on the decision to perform (or not) a resuscitation.

Reviewer 2 Report

This is a retrospective cohort study of patients sustaining an asystolic pre-hospital cardiac arrest in the Geneva area, and the factors that lead to provision of ALS or discontinuation of ALS. The results support that of existing literature in that patients who are elderly, have more co-morbidities, unwitnessed, and no bystander CPR are less likely to have an ALS attempt. The one variable that was measured that I am confused about was gender of the rescuer - what is the rationale for measuring this? Also, I think that there needs to be more detail in the discussion that this is a retrospective analysis of data and that ultimately there may be other factors involved in decision making that an only be identified through prospective and/or qualitative study.

The study period is quite historic (2009-2017) and does not reflect the current practice where there is increasing usage of TOR rules. What is the situation in Geneva in 2021 - are TORs used yet for pre-hospital situations? If so, how does this paper influence future practice? There is no mention of the 2021 ERC ALS and Ethics Guidelines which mention TORs - in particular, in particular the Ethics chapter. This needs to be included in the discussion.

Author Response

Response to Reviewer 2 Comments

Point 1: This is a retrospective cohort study of patients sustaining an asystolic pre-hospital cardiac arrest in the Geneva area, and the factors that lead to provision of ALS or discontinuation of ALS. The results support that of existing literature in that patients who are elderly, have more co-morbidities, unwitnessed, and no bystander CPR are less likely to have an ALS attempt. The one variable that was measured that I am confused about was gender of the rescuer - what is the rationale for measuring this?

Response 1: Thank you very much for this review. We decided to include the sex of the physician because recent studies have shown that male and female physicians do not always proceed the same was when treating patient (for example, the paper entitled "Association Between Patient and Physician Sex and Physician-Estimated Stroke and Bleeding Risks in Atrial Fibrillation"; DOI: 10.1016/j.cjca.2018.11.023). In our unit, a preliminary analysis of data from the study "Endotracheal Intubation Success Rate in an Urban, Supervised, Resident-Staffed Emergency Mobile System: An 11-Year Retrospective Cohort Study, DOI: 10.3390/jcm9010238)" seems to show that female physicians decide to intubate patients less often than their male colleagues. This is why we decided to include gender in the analysis.

Point 2: Also, I think that there needs to be more detail in the discussion that this is a retrospective analysis of data and that ultimately there may be other factors involved in decision making that an only be identified through prospective and/or qualitative study.

Response 2: You are perfectly right, prospective cohorts and qualitative studies could identified additional (or even better) factors to consider in these situations. We have added the following sentences (page 9) to the "limitation" paragraph of the discussion section: “In particular, the retrospective design limits the number of factors that can be studied, as only those collected for the medical record documentation could be included. Prospective studies, or even qualitative studies with residents, may help to identify additional factors influencing medical decision making.”

Point 3: The study period is quite historic (2009-2017) and does not reflect the current practice where there is increasing usage of TOR rules. What is the situation in Geneva in 2021 - are TORs used yet for pre-hospital situations? If so, how does this paper influence future practice? There is no mention of the 2021 ERC ALS and Ethics Guidelines which mention TORs - in particular, in particular the Ethics chapter. This needs to be included in the discussion.

Response 3: Thank you for this comment. We have added a paragraph (and the proposed reference) that answers these questions, following the "limitations" paragraph, in the discussion section (page 9-10). “The results of this study are in line with the new guidelines on the ethics of resuscitation and end-of-life decisions, published in 2021 [39]. Indeed, this study shows that EPs are unlikely to base their decisions solely on isolated clinical signs or isolated markers of poor prognosis. They also take into account, for example, age and comorbidities. The ToR rules, developed for emergency medical system that do not send EPs to the field, are not applied in Geneva. The 2021 guidelines on the ethics of resuscitation and end-of-life decisions recommend "that none of the existing ToR rules should be the sole determinant of when to discontinue resuscitation". In all probability, Geneva EPs take the criteria defined in the ToR rules into account when making a decision not to transport a patient with ongoing CPR. However, in this work, we studied the decision to attempt (or not) resuscitation (figure 1, decision point 2), not the decision to transport (or not) the patient with ongoing CPR (figure 1, decision point 3).”

Round 2

Reviewer 2 Report

I am happy with the authors response to my queries.